# The Mechanisms Involved in the Fluoroquinolone Resistance of *Salmonella enterica* Strains Isolated from Humans in Poland, 2018–2019: The Prediction of Antimicrobial Genes by In Silico Whole-Genome Sequencing

**DOI:** 10.3390/pathogens12020193

**Published:** 2023-01-27

**Authors:** Katarzyna Piekarska, Tomasz Wołkowicz, Katarzyna Zacharczuk, Agata Stepuch, Rafał Gierczyński

**Affiliations:** Department of Bacteriology and Biocontamination Control, National Institute of Public Health NIH—National Research Institute, 00-791 Warsaw, Poland

**Keywords:** *Salmonella enterica*, human, fluoroquinolone resistance, antimicrobial resistance genes

## Abstract

Salmonellosis remains the second most common zoonosis in Europe. Resistance to fluoroquinolones (FQs) in *Salmonella* has been increasing worldwide, with WHO considering FQ-resistant *Salmonella* spp. as high-priority pathogens. The aim of this study was a retrospective analysis of the molecular mechanisms of FQ resistance, detected among clinical ciprofloxacin-resistant *Salmonella enterica* belonging to the most common serotypes. The whole genome sequences (WGS) of tested isolates were also analysed for the occurrence of other antimicrobial resistance determinants. Out of a total of 1051 *Salmonella* collected in the years 2018–2019, 447 strains belonging to the most common serotypes in Poland were selected were screened for FQ resistance using the pefloxacin disc test according to EUCAST recommendations. All pefloxacin-resistant isolates were confirmed as ciprofloxacin-resistant using the E-test. A total of 168 (37.6%) *Salmonella enterica*, which belonged to seven serotypes, were resistant to ciprofloxacin (mostly Hadar, Virchow and Newport). A hundred randomly selected *Salmonella* were investigated by WGS. A total of 127 QRDR mutations in GyrA and ParC were identified in 93 isolates. The *qnr* genes were the only PMQR determinants detected and were found in 19% of the sequenced isolates. Moreover, 19 additional resistance genes (including: *bla,*_,_
*tet, sul, aad, aac-, ant-, aph-, flo*R, *cml*A) were identified among the FQ-resistant *Salmonella* tested that confer resistance to clinically important antibiotics such as β-lactams, tetracyclines, sulphonamides, aminoglycosides and phenicol, respectively). In conclusion, FQ resistance of human *Salmonella* in Poland is rising towards a critical level and needs to be tightly monitored.

## 1. Introduction

Non-typhoidal *Salmonella* (NTS), including *Salmonella enterica* subspecies *enterica*, which is subdivided into more than 2500 serovars based on antigenic differences in the lipopolysaccharide O antigen and two flagellin structures, is a frequent human pathogen [1]. Salmonellosis as a foodborne disease with a wide range of hosts still remains a global public health challenge. In Europe, salmonellosis is the second most common zoonosis in humans, with 87,923 confirmed human cases, as well as the most frequent cause of foodborne outbreaks accounting for 17.9% of all foodborne outbreaks reported in 2019 [2]. In the USA alone, approximately 1.2 million human cases, with 450 fatal cases, occur each year due to *Salmonella* infection [3]. Generally, NTS infections are transmitted to humans via the consumption of contaminated water and food, particularly animal-derived products. In most cases, NTS infection causes self-limiting diarrhoea. However, a life-threatening systemic infection may occur, in which case, antimicrobial treatment is required, especially in high-risk groups, such as elderly or immunocompromised patients [3]. Recently, antimicrobial resistance (AMR) among *Salmonella* human isolates, including resistance to fluoroquinolones and the third generation of cephalosporins that are the current drugs of choice for salmonellosis treatment, have been globally encountered with high frequency, also in Europe, making the empiric treatment of salmonellosis more challenging [2,4].

Although, in 2016, the World Health Organisation (WHO) listed fluoroquinolones (FQ) as a critically important class of antibacterial drugs [5], FQs have been widely used to treat various infections, including invasive non-typhoidal *Salmonella* infections in adults. Furthermore, FQs have also been used in animals to prevent, control and treat infections [6]. The widespread use of FQ to treat both humans and animals creates a selective pressure that promotes the horizontal transfer of resistance genes and the development of antimicrobial resistance, leading to the selection of resistant bacterial clones among pathogenic, commensal or even environmental bacteria [2]. FQ resistance among *Salmonella* has been on the rise worldwide for the past two decades [2,7,8,9]. In 2017, the WHO considered FQ-resistant *Salmonella* spp. as pathogens for which novel antibiotics are urgently required [10].

FQs’ main targets are two enzymes essential for bacterial replication: DNA gyrase and topoisomerase IV. In *Enterobacterales*, including *Salmonella*, quinolone resistance typically develops from the accumulation of chromosomal mutations in the quinolone resistance-determining region (QRDR) of genes encoding the subunits of DNA gyrase (GyrA, GyrB) and topoisomerase IV (ParC, ParE), primarily in GyrA and ParC [8,11,12]. Additionally, since the late 1990s, three types of plasmid-mediated quinolone resistance (PMQR) mechanisms have been identified: Qnr, which protects target enzymes; AAC(6’)-Ib-cr, which mediates the acetylation of certain quinolones; QepA and OqxAB, which produce mobile efflux pumps [13]. Although PMQRs alone are considered factors that can provide only a low level of resistance, their presence may aid bacteria to survive under low FQ pressure sufficiently to develop mutations in the DNA gyrase and/or topoisomerase IV genes, and to acquire high-level resistance to FQs [14]. These molecular mechanisms can be accumulative and together influence the observed FQ minimum inhibitory concentration (MIC) value. The genotypic features of ciprofloxacin non-susceptibility caused by the QRDR mutations and the PMQR mechanisms among NTS human isolates can vary in time and countries, but most frequently are attributable to the GyrA mutations in 83 and/or 87 codon and/or the ParC substitution in 80 codon and PMQR presence, mainly Qnr [14].

In Poland, generally, about 8000 confirmed salmonellosis cases are detected annually (ranging from 7315 in 2013 to 9718 in 2016) [15]. Furthermore, there is a lack of specific information on AMR incidence in *Salmonella* because of the insufficient number of isolates tested for phenotypic and genetic antimicrobial resistance by Polish health authorities. Therefore, it is difficult to understand the real level of antimicrobial resistance in *Salmonella*. Until now, only limited data have been available with regard to the distribution and molecular characteristics of FQ-resistant *Salmonella enterica* isolated from humans. Recently, we reported on the antimicrobial resistance of *S*. Kentucky ST 198 [16], one of the most resistant serovars occurring in humans and animals, respectively. Thus, the objective of this study was to investigate the molecular mechanisms of FQ resistance, including mutations in the QRDR (*gyr*A, *gyr*B and *par*C, *par*E) and PMQR determinants (*qnr*A, *qnr*B, *qnr*S, *qnr*C, *qnr*D, *aac(6’)-Ib*-cr, *qep*A and *oqx*AB) in clinical ciprofloxacin-resistant *Salmonella enterica* belonging to the most prevalent serotypes in Poland. Additionally, genome sequences of the tested ciprofloxacin-resistant *Salmonella enterica* isolates determined by WGS were also analysed for the presence of other AMR determinants (including: *bla,*_,_
*tet, sul, aad, aac-, ant-, aph-, flo*R, *cml*A that confer resistance to clinically important antibiotics as β-lactams, tetracyclines, sulphonamides, aminoglycosides and phenicol, respectively).

## 2. Materials and Methods

### 2.1. Salmonella enterica Isolates and Fluoroquinolone Susceptibility Testing

A total of 1051 *Salmonella enterica* strains encompassing 43 serotypes isolated from humans (patients, people from contact, convalescents and carriers) were collected in the National Institute of Public Health National Institute of Health–National Research Institute, Warsaw, Poland, in the years 2018–2019. The strains were collected as part of the National Health Program task 2016–2020 “Program for verification of accuracy of human *Salmonella* strains serotyping in Poland.” The *Salmonella* strains were identified previously by 14 Provincial Sanitary and Epidemiological Stations in Poland. All the isolates tested were re-serotyped in our laboratory by slide-agglutination methods according to the White–Kauffmann–Le Minor scheme [1] using commercial sera manufactured by: Statens Serum Institut (Denmark), Biomed (Poland) and Immunolab (Poland). Out of 1051 *Salmonella enterica* collected, 447 strains belonging to the most common serotypes in Poland were selected for FQ resistance screening tests (133 strains from 2018 and 314 strains from 2019). The isolates tested belonged to the following serotypes: Enteritidis (*n* = 222), Typhimurium (*n* = 85), Infantis (*n* = 58), monophasic Typhimurium 1,4,[5],12:i:- (*n* = 36), Kentucky (*n* = 18) published previously [16], Hadar (*n* = 13), Agona (*n* = 4), Newport (*n* = 7) and Virchow (*n* = 4). Pefloxacin (5 µg) screening test was the first criterion for selecting a strain for further study. All the pefloxacin-resistant *Salmonella* strains detected (zone diameter < 24 mm in accordance with EUCAST criteria; (http://eucast.org (accessed on 5 December 2022)) were confirmed to be resistant to ciprofloxacin by the MIC determination using the E-test strips with an antibiotic concentration gradient (bioMerieux, Marcy l’Etoile, France) according to the manufacturer’s instructions. The MIC results for ciprofloxacin were interpreted using the European Committee on Antimicrobial Susceptibility Testing (EUCAST) (http://eucast.org, accessed on 5 December 2022) criteria, while isolates with MIC > 0.06 mg/L were classified as resistant to ciprofloxacin.

### 2.2. Whole-Genome Sequencing Analysis

One hundred *Salmonella enterica* strains were randomly selected for WGS belonged to the Enteritidis (*n* = 32), Typhimurium (*n* = 12), Infantis (*n* = 30), 1,4,[5],12:i:- (*n* = 6), Hadar (*n* = 12), Newport (*n* = 5) and Virchow (*n* = 3) serotypes with ciprofloxacin MIC > 0.06 mg/L were subjected to further investigation using the whole genome sequence technique. Genomic DNA was isolated from overnight culture in LB based on a manual in-house procedure with the GTC (Guanidine thiocyanate), phenol extraction and precipitation step. The isolated DNA samples were quantified using BioSpectrometer (Eppendorf, Hamburg, Germany). The libraries were constructed using the Illumina DNA Prep chemistry (Illumin, San Diego, CA, USA), and then sequenced on MiSeq (Illumina) using MiSeq Reagent Kit v3.

The raw reads were assembled using CLC Genomics Workbench 22 (Qiagen, Hilden, Germany). *Salmonella* serotypes were confirmed using the SeqSero tool (CGE) [17]. MLST was assigned based on the raw reads using the MLST 2.0 tool (CGE; database version 22 March 2021) [18].

The acquired antimicrobial resistance genes and chromosomal point mutations were assigned from the raw reads using ResFinder 4.1 and PointFinder (CGE; database from 3 May 2022) [19]. Additionally, all the sequences were submitted to Enterobase, and the results were confirmed using an automatic analytical pipeline. The cgMLST and wgMLST (Multilocus Sequence Typing based on core and whole genome loci) variants were assigned, and specific MLST dendrograms were built. A whole genome SNP analysis was additionally prepared for each *Salmonella* serotype, and dendrograms were created using the CSI Phylogeny tool [20].

The results and dendrograms were transferred to Microreact (www.microreact.org, accessed on 5 December 2022), where a proper project was established and is available at: https://microreact.org/project/fMudrjXLMz9vzs8tDg5vK1-npz-salmonella-2018-19-publikacja, accessed on 5 December 2022. 

## 3. Results

### 3.1. Salmonella Strains and Phenotypic Resistance to Fluoroquinolones

A total of 168 (37.6%) *Salmonella enterica* isolates resistant to ciprofloxacin were obtained after screening with pefloxacin (5 µg) disc diffusion method out of 447 human isolates–133 (39.8%) in 2018 and 314 (41.4%) in 2019, respectively. The FQ-resistant *Salmonella* strains were sent from different regions of Poland and belonged to seven serotypes, as shown in Figure 1. The number of tested *Salmonella* strains, the percentage of resistant strains within each serotype and the ciprofloxacin MIC range among the seven serotypes identified in our set of ciprofloxacin-resistant isolates (*n* = 168) are shown in Table 1.

The highest percentage of ciprofloxacin-resistant strains was found among *Salmonella* Hadar, *S*. Virchow and *S*. Newport. An equally high percentage of resistant strains was found in *S*. Infantis and *S*. Enteritidis.

### 3.2. Genotypic Antimicrobial Analysis In Silico

#### 3.2.1. Resistance to Fluoroquinolones–QRDR Mutations and PMQR Genes

All the whole genome sequences obtained were entered in Enterobase under accession numbers SAL_LB2507AA–SAL_LB2553AA, SAL_LB2714AA, SAL_LB2752AA–SAL_LB2786AA, SAL_LB2788AA–SAL_LB2793AA and SAL_LB3141–SAL_LB3158AA.

The summary of the WGS results and antimicrobial resistance genes detected among ciprofloxacin-resistant *Salmonella enterica* strains is shown in Table 2. Additionally, Appendix A provide such information as the year of isolation, patient status, ciprofloxacin MIC value, and WGS results including MLST type and resistance genotype for each human isolate tested.

In 93 isolates of 100 *Salmonella enterica* analysed by means of WGS, a total of 127 QRDR mutations were identified, including 80 mutations in GyrA and 47 in ParC. Only silent mutations were found at the QRDR region of *gyr*B and *par*E. The most commonly detected mutations in GyrA were Ser83→Tyr, followed by Asp87→Tyr and Asp87→Asn. In the ParC subunit, substitution was only identified in one codon 57 (Thr→Ser), which was detected in 47 Salmonella isolates (Table 2).

Sixty (60%) of one hundred *Salmonella* isolates analysed by WGS carried at least one substitution in GyrA or ParC (Appendix A). Thirteen isolates (nine *S*. Hadar and four *S*. Newport, respectively), had amino acid substitutions in the *par*C QRDR only. Two substitutions (in GyrA and ParC) were detected in three *S*. Hadar, thirty *S*. Infantis and one *S*. Newport. The distribution of the GyrA and ParC substitutions and PMQR determinants among the seven serotypes of ciprofloxacin-resistant *Salmonella enterica* isolates are shown in Table 3. No substitution was found in six *Salmonella* isolates (Typhimurium, *n* = 1; 1,4,[5],12:i:-, *n* = 4; Enteritidis, *n* = 1) that carried *qnr*B only (Appendix A).

The PMQR determinants were detected in 19 (19%) of 100 *Salmonella enterica* isolates. The *qnr* genes were the only PMQR determinant detected. The *qnr*B19 variant was identified more frequently (in 17 isolates) than other variants: *qnr*B36 (*n* = 7); *qnr*B82 (*n* = 4) and *qnr*B67 (*n* = 2) (Table 2 and Table 3). Three and one of *S*. Hadar isolates carried a combination of three and four *qnr*B gene variants, respectively (Table 3 and Appendix A). Moreover, 7 of 12 *S*. Hadar isolates and 4 of 5 *S*. Newport carried the *qnr*B gene variants in different combinations and an amino acid substitution in 57 codons of ParC but did not contain any substitution in 83 and/or 87 codons of GyrA. Two isolates of *S*. Infantis carried *qnr*S1. Other PMQR determinants, such as *qnr*A, *qnr*C, *qnr*D, *aac(6’)-Ib*-cr, *qep*A and *oqx*AB, were not detected among the *Salmonella enterica* isolates tested.

#### 3.2.2. In Silico Antimicrobial Resistance Genes Detected among FQ-Resistant Salmonella Isolates

The WGS results for *Salmonella* isolates resistant to fluoroquinolones are shown in the Appendix A. In addition to the PMQRs, a total of 19 resistance genes were identified among the FQ-resistant *Salmonella* tested. These genes were predicted to confer resistance to clinically important classes of antibiotics other than FQs (Table 2 and Appendix A). A list of the resistance genes and their prevalence among *Salmonella* isolates is shown in Table 2. The distribution of antimicrobial resistance genes among the seven *Salmonella enterica* serotypes is shown schematically in Figure 2.

A total of five genes encoding β-lactamases were identified among the tested *S. enterica*, with the most common *bla*_TEM-1_ and *bla*_CARB-2_ alleles among *S*. Typhimurium (Table 2 and Appendix A). Moreover, two *S*. Typhimurium isolates carried *bla*_CMY-2_, while one isolate had both, *bla*_CMY-2_ and *bla*_CARB-2_ (Appendix A).

Six genes encoding aminoglycoside resistance–*aac(6′)-Iaa*, *ant(3″)-Ia*, *aph(3″)-Ib*, *aph(6)-Id*, *aad*A1 and *aad*A2–were detected (Table 2). Each of the seven *Salmonella* serotypes tested had at least one aminoglycoside resistance gene (Appendix A).

Three distinct *tet* genes encoding tetracycline resistance efflux pumps were identified in the FQ-resistant *Salmonella* isolates tested. The *tet*A was the most common gene detected in 11 of 12 *S*. Hadar, 29 of 30 *S*. Infantis and 3 of 5 *S*. Newport. Additionally, 9 of 12 *S*. Typhimurium isolates and 4 of 6 monophasic 1,4,[5],12:i:- carried the *tet*G and *tet*B genes, respectively.

Sulphonamide resistance was predominantly encoded by three alleles–*sul*1, *sul*2 and *sul*3. The sul1 dominated in *S. Typhimurium* and *S. Infantis*, while *sul*2–in monophasic 1,4,[5],12:i:- *S. Typhimurium* (Figure 2, Appendix A).

Phenicol resistance in the *Salmonella* isolates tested was encoded by two genes–*flo*R and *cml*A1 detected among 10 (9 *S. Typhimurium* and 1 *S. Infantis*) and two monophasic *S. Typhimurium* (Table 2 and Appendix A).

## 4. Discussion

There is an urgent need to preserve the effectiveness of antimicrobials for humans and animals, with the “One-health” initiative seeking to promote the responsible use of antimicrobials. Antimicrobial agents used in food-producing animals and in human medicine in Europe are frequently the same or belong to the same classes, and their use in both humans and animals might result in the selection of resistant clones, and the development of AMR [2]. Moreover, bacterial resistance to antimicrobials occurring in food-producing animals can spread to humans via food-borne routes, as has been observed for such zoonotic bacteria as *Salmonella*, by environmental contamination or through direct animal contact [2]. It is worth noting that, in accordance with Decision 2018/945/EU [21] on communicable diseases and related special health issues, antimicrobial resistance among *Salmonella* isolates in humans should be monitored. Furthermore, the WHO stated that the transmission of bacterial infections from non-human sources to humans, with disease-causing potential, is more evident in some bacteria (including non-typhoidal *Salmonella*) and the potential for such transmission should be recognised [5]. Nowadays, it is a well-recognised and generally acknowledged issue that fluoroquinolones are extensively used in medicine, veterinary and animal production [22,23].

In this study, we described the prevalence of FQ resistance among seven *Salmonella enterica* serotypes commonly occurring in humans in Poland. The 447 clinical human *Salmonella enterica* from different regions of Poland isolated in the years 2018–2019 were screened for FQ resistance. Our results indicated that there was a high level (37.6%) of FQ resistance among the isolates tested. As shown in Table 1, FQ resistance was most frequently detected among *S. Hadar* (92.3%), *S. Virchow* (75%) and *S. Newport* (71.4%). This was followed by *S. Infantis* (55.2%), *S. Enteritidis* (44.1%), monophasic 1,4,[5],12:i:- (16.2%) or *S. Typhimurium* (14.3%). Their ciprofloxacin MICs ranged from 0.125 to 3 mg/L. In accordance with the EFSA and ECDC joint report on antimicrobial resistance, in total 13.5% of *Salmonella* spp. isolated from humans were ciprofloxacin-resistant in Europe. Among them were: 20.9% Enteritidis, 19.8% Infantis, 5.6% monophasic Typhimurium and 5.1% Typhimurium, respectively [2].

Typically, chromosomal mutations inside the QRDRs of genes encoding DNA gyrase and topoisomerase IV, mainly the GyrA and/or ParC subunits, or the accumulation of mutations in these subunits, were considered to be important for ciprofloxacin resistance [8,11]. Out of the 100 *Salmonella* isolates resistant to ciprofloxacin (MIC > 0.06 mg/L) analysed by means of WGS in this study, 80 isolates had mutations in the *gyr*A QRDR, including 45 at Ser-83 and 35 at Asp-87. Both mutations, in GyrA and ParC, have been found among 34 *Salmonella* isolates tested. Notably, 30 *S. Infantis* with ciprofloxacin MIC ≥0.25 mg/L had an amino acid substitution in GyrA at codon 83 Ser→Tyr (*n* = 12) or codon 87 Asp→Tyr (*n* = 18) and a simultaneous alteration of ParC at codon 57 (Thr→Ser). Moreover, mutation Thr57→Ser in ParC was the only substitution detected among 47 tested *Salmonella* strains with ciprofloxacin MIC range 0.125–0.25 mg/L in *S. Hadar*, (0.19–0.75 mg/L) in *S. Newport* and (0.25–3 mg/L) in *S. Infantis*, respectively. This is in line with the reports of other authors that found Thr57Ser as the most prevalent mutation in ParC in NTS human isolates [14,24,25]. In accordance with a recent study, the ParC mutation at Thr57→Ser was detected as the main substitution in *Salmonella enterica* isolates from food-producing animals in China with a broad range of the ciprofloxacin MIC value (0.008–64 mg/L) [26]. Although authors suggest that the impact of the Thr57Ser mutation alone on quinolone susceptibility must be low, it may affect quinolone susceptibility when combined with the GyrA substitutions [26].

The plasmid-mediated quinolone resistance (PMQR) determinants are considered the cause of decreased susceptibility to FQs and factors facilitating the selection and acquisition of high-level FQ resistance [27]. In this study, *qnr* were the most prevalent PMQR genes detected among 7 of 12 *S. Hadar*, 4 of 5 *S. Newport* and 4 of 6 monophasic *S. Typhimurium*, 2 of 30 *S. Infantis*, 1 of 12 *S. Typhimurium* and 1 of 32 *S. Enteritidis* tested with ciprofloxacin MIC ≥0.19 mg/L, respectively. *qnr*B (variant: q*nr*B19, *qnr*B36, *qnr*B67 and *qnr*B82) was found in 17 isolates belonging to the serotypes Typhimurium (1/12), monophasic Typhimurium (4/6), Hadar (7/12), Newport (4/5), Enteritidis (1/32), while *qnr*S1 was found in 2 of 30 isolates of Infantis. Several studies have globally reported an association between the PMQR genes and the ciprofloxacin non-susceptible NTS isolates [14,25,28,29,30,31]. Additionally, the QRDR mutations and PMQR genes in NTS human isolates can be country specific. Similar to other reports, we detected NTS, including four monophasic Typhimurium, one Typhimurium and Enteritidis isolates, harbouring only PMQR genes without mutations in QRDR [14]. Interestingly, we identified three NTS strains (Hadar, Newport, monophasic Typhimurium 1,4,[5],12:i:-), three Hadar strains and one Newport harbouring two, three and four different variants of *qnr*B, respectively.

In addition to the variety of existing FQ resistance mechanisms, we found a high diversity of resistance genes by means of the in silico analysis of *Salmonella* WGS. Resistance to aminoglycosides was most common and showed the highest diversity (*n* = 6), including the aminoglycoside N-acetylotransferases, O-nucleotydylotransferases and O-phosphotransferases groups that modify aminoglycosides. In our study, *aac(6′)-Iaa* was the most frequently identified gene, in 68% (68/100) *Salmonella* isolates analysed by means of WGS. Less commonly found genes were: *ant(3″)-Ia* (*n* = 29) *aph(3″)-Ib* (*n* = 19) and *aph(6)-Id* (*n* = 18). The *aad*A1 and *aad*A2 genes also detected in our study confer aminoglycoside resistance, more specifically streptomycin and spectinomycin resistance. These resistance genes can be carried on mobile genetic elements that facilitate transfer between bacteria and promote dissemination within bacterial populations [32]. This finding corroborates several reports demonstrating the presence of multiple aminoglycoside resistance genes in *Salmonella* serovars [33,34]. Moreover, the “One-health” results from Canadian researchers [34] suggested that the use of lincomycin-spectinomycin in chickens may be selecting for gentamicin-resistant *Salmonella* in broilers, and then these resistant strains may be acquired by humans.

The resistance genes *tet* and *sul* that may confer resistance against tetracycline and sulphonamide, respectively, were also highly frequent among the WGS analysed *Salmonella* isolates, with the frequency amounting to 56% and 49%, respectively. Among them, tetA (*n* = 43) and sul1 (*n* = 39) were the most common. Based on the EFSA and ECDC joint report on antimicrobial resistance, in total 29% and 25.6% of *Salmonella* spp. isolated from humans in Europe are resistant to sulphonamides and tetracyclines, respectively [2]. The report also points to the high-level sulphamethoxazole resistance noted in broilers and turkeys (33.9% and 13.7%, respectively), and high-level tetracycline resistance (35.5% in broilers and 57.3% in turkeys, respectively) [2]. This may indicate that the indiscriminate use of antimicrobials in animal husbandry is also causing an increase in the antimicrobial resistance in *Salmonella* isolated from humans [35].

According to the ECDC and EFSA joint report [2], in 2019 in Europe, a high (25.8%) percentage of human *Salmonella* isolates were resistant to ampicillin, including monophasic Typhimurium (87.1%), Typhimurium (54.3%), Infantis (18.3%) and Enteritidis (8.1%). In our study, the *bla*_TEM-1_ gene encoding ampicillin resistance was detected in twelve (12%) Salmonella isolates belonging to monophasic 1,4,[5],12:i:- (*n* = 5), Infantis (*n* = 3), Newport (*n* = 2) and Typhimurium (*n* = 2). Interestingly, 8 isolates of 12 S. Typhimurium had the *bla*_CARB-2_ gene conferring resistance to ampicillin. The occurrence of the *bla*_CARB-2_ gene, earlier identified as *bla*_PSE-1_, in *Salmonella* has been limited to a few reports that concerned isolates from humans, animals and animal-derived products [36,37,38]. It is important to note that two *S*. Typhimurium isolates additionally carried *bla*_CMY-2_ (the acquired AmpC β-lactamase gene) that mediates resistance to a wide variety of beta-lactams, particularly cephalosporins. Our result is in line with the aforementioned ECDC and EFSA joint report where AmpC was identified in 0.1% of all the human *Salmonella* isolates tested [2].

## 5. Conclusions

In conclusion, our study demonstrated that a significant percentage of human *Salmonella* isolates is resistant to FQs. Although a small number of *Salmonella* strains was tested on the national scale, the results obtained illustrate the FQ resistance situation in Poland. The development of nontyphoidal *Salmonella* isolates resistant to FQs is of clinical importance since it may be associated with reduced effectiveness of therapies and represents a substantial public health concern. Zoonotic *Salmonella* is the leading foodborne pathogen, with global travel and food import significantly increasing the risk of acquiring salmonellosis. Moreover, our study strongly supports the need to carry out the surveillance and monitoring of AMR, including FQ resistance among human *Salmonella* isolates. In addition, in accordance with the “One Health” concept, collaboration and integrated surveillance for zoonotic *Salmonella*, including isolates originating from humans and food-producing animals, is necessary to elucidate trends in antimicrobial resistance. Additionally, it is worth remembering the role of the rational and prudent use of antibiotics including FQs, both in humans and animals. Because, as the Australian case indicated, banning the use of FQs in food-producing animals correlated with reduced FQ resistance in bacteria isolated from food, animals and patients [39].

## Figures and Tables

**Figure 1 pathogens-12-00193-f001:**
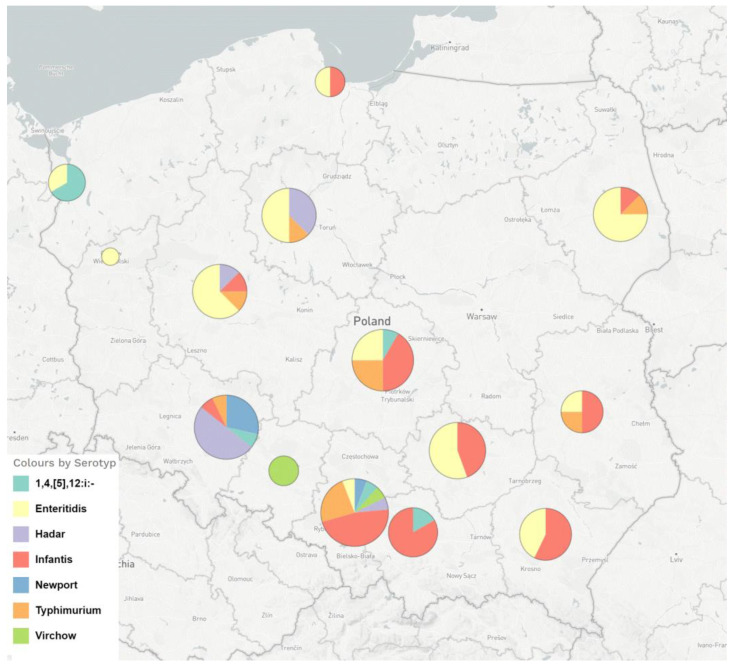
The geographical distribution and serotypes of FQ-resistant *Salmonella enterica* strains isolated from humans in Poland in the years 2018–2019 and analysed by WGS.

**Figure 2 pathogens-12-00193-f002:**
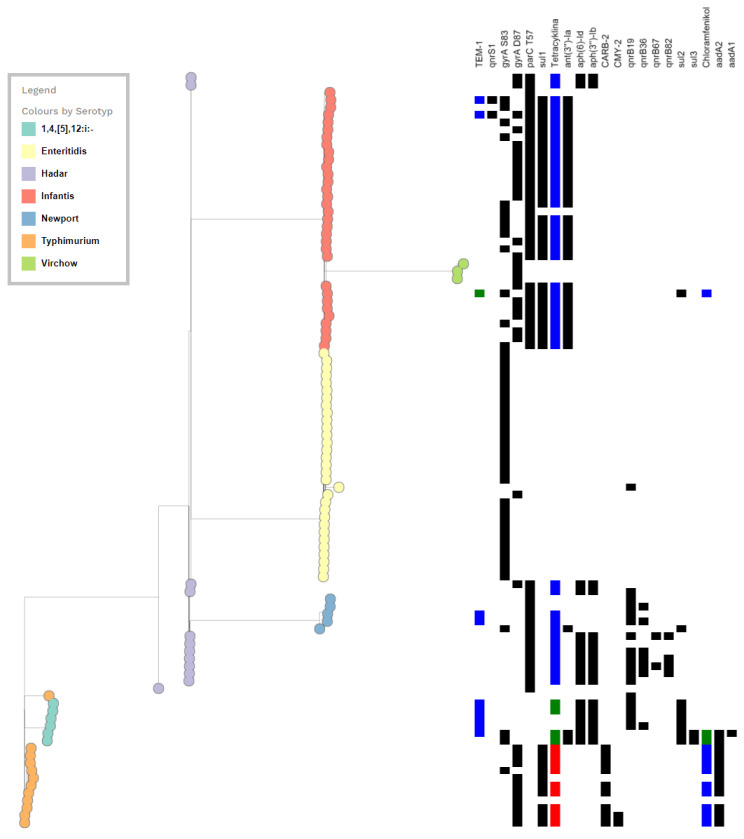
Schematic diagram of the distribution of antimicrobial resistance genes among the 7 *Salmonella enterica* serotypes.

**Table 1 pathogens-12-00193-t001:** The number of tested *Salmonella* serotypes, the number of ciprofloxacin-resistant strains and the ciprofloxacin MIC range for strains isolated from humans in Poland.

Serotype	No of Isolates Tested	No of Ciprofloxacin Resistant Isolates (%)	Ciprofloxacin MIC Value (mg/L)
1,4,[5],12:i:-	37	6 (16.2%)	0.38–0.5
Hadar	13	12 (92.3%)	0.125–0.25
Infantis	58	32 (55.2%)	0.19–3
Newport	7	5 (71.4%)	0.19–0.75
Typhimurium	84	12 (14.3%)	0.125–0.38
Virchow	4	3 (75%)	0.125–0.25
Enteritidis	222	98 (44.1%)	0.125–0.75

Ciprofloxacin MIC interpretation according to the EUCAST clinical breakpoint Table 2022 (mg/L): S ≤ 0.06; R > 0.06.

**Table 2 pathogens-12-00193-t002:** The total number of QRDR mutations and antimicrobial resistance genes including PMQR among the 100 clinical ciprofloxacin-resistant *Salmonella enterica* isolates of 7 serotypes.

Antimicrobial Class	AMR Gene			Isolates Positive for Each Gene (*n*)
Fluoroquinolones	QRDR mutations	GyrA	Ser83→Tyr	45
			Asp87→Asn Asp87→Tyr	14 21
		ParC	Thr57→Ser	47
	PMQRs	*qnr*B19		17
		*qnr*B36		7
		*qnr*B67		2
		*qnr*B82		4
		*qnr*S1		2
β-lactams	*bla* _TEM-1A_			2
	*bla* _TEM-1B_			9
	*bla* _TEM-1C_			1
	*bla* _CARB-2_			8
	*bla* _CMY-2_			2
Tetracyclines	*tet*(A)			43
	*tet*(B)			4
	*tet*(G)			9
Sulphonamides	*sul*1			39
	*sul*2			8
	*sul*3			2
Aminoglycosides	*aac(6′)-Iaa*			68
	*ant(3″)-Ia*			29
	*aph(3″)-Ib*			19
	*aph(6)-Id*			18
	*aad*A1			1
	*aad*A2			12
Phenicol	*flo*R			10
	*cml*A1			2

**Table 3 pathogens-12-00193-t003:** The distribution of the GyrA and ParC substitutions and PMQR determinants among the 7 serotypes of ciprofloxacin-resistant *Salmonella enterica* isolates.

Serotype (Isolates taken into the WGS)	GyrA	No Isolates	ParC	No Isolates	PMQR			
					*qnr*B	No Isolates	*qnr*S	No Isolates
Typhimurium (*n* = 12)	Ser83→Tyr	1	-	-	*qnr*B 19	1	-	-
	Asp87→Asn	10						
1,4,[5],12:i:- (*n* = 6)	Ser83→Tyr	2	-	-	*qnr*B 19 *qnr*B 36	4 1	-	-
Hadar (*n* = 12)	Asp87→Asn	2	Thr57→Ser	12	*qnr*B 19 *qnr*B 36 *qnr*B 67 *qnr*B 82	7 4 2 4	-	-
	Asp87→Tyr	1						
Infantis (*n* = 30)	Ser83→Tyr	12	Thr57→Ser	30	-	-	*qnr*S 1	2
	Asp87→Tyr	18						
Newport (*n* = 5)	Ser83→Tyr	1	Thr57→Ser	5	*qnr*B 19 *qnr*B 36	4 2	-	-
Virchow (*n* = 3)	Asp87→Asn	2	-	-	-	-	-	-
	Asp87→Tyr	1						
Enteritidis (*n* = 32)	Ser83→Tyr	29	-	-	*qnr*B 19	1	-	-
	Asp87→Tyr	1						

## Data Availability

Not applicable.

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
