# Peer review of "The Mechanisms Involved in the Fluoroquinolone Resistance of Salmonella enterica Strains Isolated from Humans in Poland, 2018–2019: The Prediction of Antimicrobial Genes by In Silico Whole-Genome Sequencing"

_pathogens, 2023, doi:10.3390/pathogens12020193_

Round 1

Reviewer 1 Report

The study by Piekarska et al has addressed a major health concern of antimicrobial resistance in Salmonella isolated from Poland. The study is meticulously carried out and strains are rigorously tested for resistance. Resistance phenotypes seen experimentally are well supported by WGS analysis. The study is also extensive in regards to the broad range of serotypes tested and not necessarily restricted to a particular serotype. Some of my minor comments to help improve the manuscript are listed below:

1. Line 14 - enterica should be italicized.

2. The authors should clarify why 447/1051 isolates were screened for FQ resistance.

3. Clarification is required for the use of pefloxacin disc test instead of ciprofloxacin.

4. Lines 22-23 in the abstract should state the additional resistance genes and the clinically important antibiotics. 

5. Line 25 - Remove the word surveillance.

6. Line 32 - Cite a reference for the classification system and include the LPS and flagellin gene names used for classification. 

7. Line 69 - Cite a reference.

8. Line 84 - Change to "molecular mechanisms of FQ resistance".

9. Line 89 - State the other AMR determinants. 

10. Line 94 - Correct NIH to National Institutes of Health

11. Line 109 - Cite a reference.

12. Line 120 - What is GTC?

13. Line 132 - Describe what is cgMLST and wgMLST.

14. Table 2 - Correct b-lactems to b-lactams

15. Line 199 - Italicize in silico and Salmonella. 

16. Line 218 - Italicize tet.

17. Provide High-resolution figure 2

Author Response

The authors are very grateful for valuable comments.

1. Line 14 - enterica should be italicized. Corrected

2. The authors should clarify why 447/1051 isolates were screened for FQ resistance. Corrected

3. Clarification is required for the use of pefloxacin disc test instead of ciprofloxacin. Corrected

4. Lines 22-23 in the abstract should state the additional resistance genes and the clinically important antibiotics. Corrected

5. Line 25 - Remove the word surveillance. Corrected

6. Line 32 - Cite a reference for the classification system and include the LPS and flagellin gene names used for classification. Corrected

7. Line 69 - Cite a reference. Corrected

8. Line 84 - Change to "molecular mechanisms of FQ resistance". Corrected

9. Line 89 - State the other AMR determinants. Corrected

10. Line 94 - Correct NIH to National Institutes of Health Corrected

11. Line 109 - Cite a reference. Corrected

12. Line 120 - What is GTC? Corrected

13. Line 132 - Describe what is cgMLST and wgMLST. Corrected

14. Table 2 - Correct b-lactems to b-lactams Corrected

15. Line 199 - Italicize in silico and Salmonella.  Corrected

16. Line 218 - Italicize tet. Corrected

17. Provide High-resolution figure 2 Corrected

Reviewer 2 Report

Overall, the study is well designed and the manuscript is well written.

Author Response

The authors are very grateful for manuscript review.

Reviewer 3 Report

This study entitled " The mechanisms involved in the fluoroquinolone resistance of Salmonella enterica strains isolated from humans in Poland, 2018-2019: the prediction of antimicrobial genes by in silico whole-genome sequencing", has discussed an important topic and was carried out to investigate the molecular FQ mechanisms, including mutations in the QRDR (gyrA, gyrB and parC, parE) and PMQR determinants (qnrA, qnrB, qnrS, qnrC, qnrD,  aac(6')-Ib-cr, qepA and oqxAB) in clinical ciprofloxacin-resistant Salmonella enterica belonging to the most prevalent serotypes in Poland. Also, genome sequences of the tested ciprofloxacin-resistant Salmonella enterica isolates determined by WGS were also analysed for the presence of other AMR determinants. However, there are some major points that should be addressed before the paper can be accepted for publication:

-Regarding the 100 isolates selected for whole-genome sequencing, the authors in the abstract mentioned that these isolates were "randomly" selected; however, in methodology, they mentioned that isolates with MIC >0.06 are only exposed to WGS. This means that only 100 isolates were resistant to ciprofloxacin as determined by E-test strip. This point should be clarified, as also the results showed that 168 isolates were resistant to ciprofloxacin.

-In methodology, the sequences were analyzed using the MLST 2.0 tool (CGE), however,no MLST results were determined for the sequenced isolates.

-Lines 161-163: It is obvious that only 5 strains with the following accession numbers (SAL_LB2507AA-SAL_LB2553AA, SAL_LB2714AA, SAL_LB2752AA-SAL_LB2786AA, SAL_LB2788AA-SAL_LB2793AA and SAL_LB3141-SAL_LB3158AA) were imported into Enterobase, and subjected to analysis. However, 100 isolates were sequenced, so what about the other 95 sequenced isolates?

Author Response

The authors are very grateful for valuable comments.

Regarding the 100 isolates selected for whole-genome sequencing, the authors in the abstract mentioned that these isolates were "randomly" selected; however, in methodology, they mentioned that isolates with MIC >0.06 are only exposed to WGS. This means that only 100 isolates were resistant to ciprofloxacin as determined by E-test strip. This point should be clarified, as also the results showed that 168 isolates were resistant to ciprofloxacin. Corrected

-In methodology, the sequences were analyzed using the MLST 2.0 tool (CGE), however,no MLST results were determined for the sequenced isolates.  

  • MLST results are listed in the supplementary materials (Table S4-S10) and in the microreact project (link in the text). Although we agree that it is a good practice to checked the MLST, the STs are not mentioned and added in the text because it is well known that in Salmonella STs are serotype specific.

-Lines 161-163: It is obvious that only 5 strains with the following accession numbers (SAL_LB2507AA-SAL_LB2553AA, SAL_LB2714AA, SAL_LB2752AA-SAL_LB2786AA, SAL_LB2788AA-SAL_LB2793AA and SAL_LB3141-SAL_LB3158AA) were imported into Enterobase, and subjected to analysis. However, 100 isolates were sequenced, so what about the other 95 sequenced isolates?

  • Everything is ok. These are accession numbers ranges and that is why there are dashes between the numbers. Just to clarify that we have added spaces before and after dashes.

Round 2

Reviewer 3 Report

Comments are addressed